# Assessment of Oral Skills in Adolescents

**DOI:** 10.3390/children8121136

**Published:** 2021-12-04

**Authors:** Marta Gràcia, Jesús M. Alvarado, Silvia Nieva

**Affiliations:** 1Communication, Oral Language and Diversity Research Group (CLOD), Department of Cognition, Development and Psychology of Development, Faculty of Psychology, University of Barcelona, 08035 Barcelona, Spain; mgraciag@ub.edu; 2Cognitive Psychology, Measurement and Modelisation of Processes Research Team, Department of Psychobiology and Behavioral Sciences Methods, Faculty of Psychology, Campus of Somosaguas, 28223 Madrid, Spain; 3Cognitive Psychology, Measurement and Modelisation of Processes Research Team, Department of Experimental Psychology, Cognitive Processes and Speech and Language Therapy, Faculty of Psychology, Campus of Somosaguas, 28223 Madrid, Spain; silnieva@ucm.es

**Keywords:** oral skills, oral competence, self-report, middle school, assessment model

## Abstract

There is broad consensus on the need to foster oral skills in middle school due to their inherent importance and because they serve as a tool for learning and acquiring other competences. In order to facilitate the assessment of communicative competence, we hereby propose a model which establishes five key dimensions for effective oral communication: interaction management; multimodality and prosody; textual coherence and cohesion; argumentative strategies; and lexicon and terminology. Based on this model, we developed indicators to measure the proposed dimensions, thus generating a self-report tool to assess oral communication in middle school. Following an initial study conducted with 168 students (mean age = 12.47 years, SD = 0.41), we selected 22 items with the highest discriminant power, while in a second study carried out with a sample of 960 students (mean age 14.11 years, SD = 0.97), we obtained evidence concerning factorial validity and the relationships between oral skills, emotional intelligence and metacognitive strategies related to metacomprehension. We concluded that the proposed model and its derived measure constitute an instrument with good psychometric properties for a reliable and valid assessment of students’ oral competence in middle school.

## 1. Introduction

### 1.1. Oral Communicative Skills

Research on oral language learning has yielded a variety of theoretical models which attribute greater or lesser explanatory power to the influence of the environment (interlocutors) on oral language learning [1,2,3]. Studies conducted from a linguistic perspective traditionally focused on formal aspects related to grammar (phonology, morphology and syntax); however, increasing research is now being carried out on the pragmatics of communication [4,5]. Analyses of oral language corpora and the application of psycholinguistic approaches, drawing on emergentist and/or learning-based theories, broadened the scope of oral communicative competence research to include a functional perspective within natural settings [6,7,8]. However, most studies on language acquisition and communicative development focused on understanding developmental trajectories, milestones and explanatory mechanisms in the child population.

Meanwhile, research on interaction situations that facilitate the development of oral skills in school contexts generated an abundant literature illustrating a variety of approaches, including language teaching, dialogic teaching, content and language integrated learning (CLIL) programs and integrated language teaching [9]. Some authors created specific teacher development tools to promote such methods [10,11,12,13,14].

However, although studies on dialogic teaching, the conversation method and integrated language teaching programs made a significant and proven contribution, as did proposals for teacher development, to acquiring the competences necessary to implement these approaches, such research again focused mainly on pre-school and primary education. Evidence regarding the adolescent population is scarce [15,16,17,18] and is currently focused on language teaching [19], literacy, or language and communication disorders [20]. There is also a research interest in relating language competence to aspects such as victimization [21].

In adolescence, spoken language becomes more specialized at a semantic level, more complex at a syntactic level and more abstract. At the same time, different formal registers are introduced, while metacognition increases, enabling conscious reflection on aspects of formal language (which are also included in the school curriculum) and facilitating the use of linguistic constructions tailored to the demands of different environments and interlocutors. This is reflected in the shift from spoken to written skills, which is increasingly adapted to academic requirements. The formal language dimension was superficially studied [16] and the pragmatic dimension (relating to the use of language for communication) has yet to be explored, although there are signs in the literature that progress is being made in the assessment of spoken language and the extent to which it can predict cognitive skills [22].

Written from a holistic, learner-centered perspective, the chapter on communication in the International Classification of Functioning, Disability and Health for Children and Youth (ICF-CY) lists receiving spoken and non-verbal messages, conversing (starting, sustaining, and ending) with one or more people, and debating as essential skills for functional performance in daily life [23]. Implementation in the field of education was affected by the Bologna Plan and the Common European Framework of Reference for Languages (CEFR), which brings together the language policies of the Council of Europe. The CEFR provides standardized guidelines for learning, teaching, and measuring levels of spoken and written comprehension, production and interaction as basic skills [24]. In Spain, the CEFR was incorporated into Organic Law 8/2013 for the Improvement of Educational Quality (Spanish initials: LOMCE) [25].

### 1.2. Self-Perceived Ability and Self-Efficacy

One method of assessing competences that is in the process of development, is to administer questionnaires measuring self-perceived ability, a method that was used tangentially with adolescents in relation to general competences [26,27]. Such self-perceptions can be measured by means of tests of self-efficacy, defined by Bandura [28] as people’s beliefs in their capabilities to achieve goals. These goals depend on context and on the performance of a particular task at a particular time and place.

Self-efficacy can be organized according to the difficulty of tasks or situations and requires knowledge of the skills necessary to performance of the tasks. It thus requires multidimensional metacognition skills that are related to self-regulation, and which predict motivation, learning and, consequently, academic performance [29,30]. The relationship between self-efficacy and academic performance was studied by means of questionnaires administered to adolescents [31].

### 1.3. Instruments for Assessing Self-Perceived Communicative Competence

Despite the importance of oral communication, to date, few instruments have been developed to assess students’ self-perceived communicative competence. The aim of the present study was to meet this need by providing Middle School (MS) students with an instrument that helps them reflect on their oral skills, which is the first step towards improving on this and other competences.

Among the few instruments developed is Demir’s proposed *Speaking Skills Self-Efficacy Scale* (SSS) [22], intended to determine students’ self-efficacy with respect to their oral expression abilities. The SSS contains 25 items scored using a Likert-type scale, and it has a Cronbach’s alpha reliability of 0.90. The items measure various effective communication strategies, for example: *I start my speech appropriately*; *I*
*change my speech according to the environment*; *I pay attention to protocols in my speech*; *I end my speech with appropriate expressions*; *I try to make my speech understandable*; *I give accurate answers to the questions addressed to me*; and *I*
*try to use the new words I have learnt in my speech*.

Demir’s scale is interesting and shares many points in common with the proposal reported in this paper, insofar as it partially addresses our study aim: to determine adolescent MS students’ perceptions of their oral competence [22]. However, we believe that a more detailed explanation of the underlying dimensions is required to ensure that the elements considered crucial to the development of students’ oral skills are included in the assessment instrument in order to determine each learner’s baseline.

Within a university education context, Campos et al. [32] administered a questionnaire intended to measure trainee teachers’ self-assessment of their oral communication competence (25 items) and their assessment of the related education received during their degree course (9 items). All items were scored using a Likert scale with five response options, except for one item with an open-ended response option. The authors based their scale on an instrument developed by Gallego and Rodríguez [33], but added items grouped into a self-exploratory dimension—competence as a sender and competence as a receiver—to explore students’ self-perceptions of their skills as trainee teachers. Their results show that competence as a sender was rated lower than competence as a receiver, and that oral expression was perceived as particularly limited in more formal communication situations. Further differences were observed in relation to variables such as gender, age, year and previous academic or professional experience, whereby the highest self-assessments were those made by men, older people, students in a higher year, participants already holding a university degree and those who combined their education with a public-facing job.

In an attempt to develop a theoretically grounded instrument, we drew on studies of oral language teaching from a pragmatic perspective [10,11,34,35] that analyzed interaction situations in higher education contexts, in which five major dimensions or content domains can be differentiated, requiring sampling in order to obtain a valid instrument.

An inductive and deductive analysis (essentially theories of argumentation and an analysis of discourse) of university classes dealing with a variety of subjects was carried out as part of two research projects focused on improving initial teacher training, using students of Initial Teaching Education (ITE) at Catalan universities [10,11,12] from a pragmatic perspective. The analysis indicates that most production can be grouped into 5 dimensions. In this work the dimensions were included in a heading that was used to jointly analyze the interaction taking place during the classes, including the production and actions of students and the teacher, in order to become aware of what could be improved and to introduce changes. Classes were understood as contexts of communicative interaction which fostered the construction of knowledge [12,14].

The first dimension, interaction management, refers to the ability to manage network interactions. It comprises four sub-dimensions: (1) the use of interactive markers that favor verbal and non-verbal network participation [36,37,38]; (2) participation in communicative turns [39]; (3) participation management; and (4) the use of politeness strategies [40,41].

This first dimension has its origin in the theories of analysis of discourse and argument [39], as well as the works on interactions in natural contexts between adults and children [12], which note the importance of encouraging adults to manage the conversation initially and then gradually cede control of the conversation to the learners.

The second dimension, multimodality and prosody, comprises two sub-dimensions: (1) the ability to implement the multimodal resources (hand, facial and body gestures) that characterize social interaction and contribute qualitatively to verbal expression [42,43,44,45]; and (2) the use of prosody, the ability to use utterances to clearly convey content and emotions (intonation in different language constructions—intonational curves and prosodic accent—and characteristics presented by the utterance, in relation to both prosodic and paralinguistic elements such as intensity, speed, pitch and articulation) [37].

With respect to the second dimension, it is worth highlighting multimodality, as it introduces psychological aspects referring to the modality of communication, which are not only oral but also gestural, using aspects such as augmentative communication systems and sign language. We understand any modality of language as something that contributes to making conversation in class the key element of learning of all the content and to the development of competences, including self-competence.

The third dimension, textual coherence, and cohesion, also comprises two sub-dimensions: (1) coherence (expounding, grouping and sequencing ideas to endow the content of the information with a sense of unity and completeness) [46,47,48]; and (2) textual cohesion (linking fragments and sentences by means of connectors and processing information using discourse markers or modal operators) [49,50,51].

An analysis of coherence and cohesion is key in any type of analysis of discourse, including the discourse which takes place in university classrooms, given that it helps future teachers be aware of the elements to take into account when structuring discourse, in this case of the conversational and interactive form.

The fourth dimension, argumentative strategies, includes the elements that refer to the ability to formulate and argue a position using reasoning, with the aim of reaching consensus. This has seven sub-dimensions: (1) formulation of a thesis; (2) validity of the arguments; (3) exposition of counterarguments; (4) identification of fallacies; (5) formulation of conclusions; (6) use of evidential constructions indicating the source of the information; and (7) use of patterns in the argumentative sequence [52,53,54,55,56,57].

Additionally, some indicators are worth highlighting in the fourth dimension such as formulating conclusions, which are a key element for teachers to be able to teach and model in the class, and a relevant competence not only for the linguistic development but also cognitive development of students. Similarly, the reference to constructing argumentative sequences, in which students’ and teachers’ arguments and counter-arguments are integrated, constitutes an important contribution with respect to instruments analyzing an interaction that is more one-way in nature, which are only focused on the teacher or students, and do not include the interactive nature of classroom situations.

The fifth and final dimension, lexicon and terminology, refers to the capacity to produce an accurate and varied expression, and has two sub-dimensions: (1) the common lexicon; and (2) domain-specific terminology [58].

With respect to the fifth dimension, we highlight the difference between the use of common vocabulary, and the use of concepts and a lexicon that must be incorporated into the content worked on in class, which the teachers must continue modeling and fostering among their students (who are set to become future teachers themselves).

Previous studies by the research team using these five dimensions to assess first-year university students [10,11], revealed their effectiveness in determining both the teacher’s and the students’ progress in oral competence during whole class discussion sessions and collaborative group discussions or conversations.

An instrument that enables the self-assessment of oral competence in these five communicative dimensions is important not only because it enables an accurate assessment of a student’s level and progress, but also because improvements in these basic skills are likely to exert an impact on other related variables, such as emotional intelligence and self-regulation in reading comprehension.

### 1.4. Relationship between Oral Skills, Emotional Intelligence and Metacognitive Strategies Applied to Reading

Efficient oral communication is related to the sensitivity to perceive others’ reactions, and must be self-regulating in order to adapt to the environment; therefore, it must involve emotional and self-regulatory elements. In addition, given that oral competence develops alongside reading skills, one might expect self-regulation in reading comprehension (metacomprehension) to be related to speaking self-efficacy. Flavell linked metacognition and metacomprehension with different dimensions of spoken and written communication comprehension and expression, as well as to executive functions, including self-regulation [59]. In relation to the link between oral skills and reading comprehension, Clarke et al. [60] found that listening comprehension is the main predictor of reading comprehension, and that this relationship increases with age. An applied example of the greater improvement in oral expression over other skills, such as reading or listening comprehension, is given by research on MS students, using micro laptops in the classroom [61].

Meanwhile, there is broad consensus regarding the importance of emotional intelligence (EI) in effective oral communication [62,63,64,65,66,67]. In addition, a very close relationship was found between leadership ability and communicative competence [64], whereby leaders with good communicative competence often had high EI. Many students report a positive relationship between communicative competence and EI [68,69]. More specifically, a study by Marzuki et al. [65] with university students showed a significant relationship between EI and information and communication technology skills.

Since any valid psychological assessment measure must demonstrate construct validity by showing evidence of a convergent relationship with other theoretically related variables, instruments that measured EI and metacomprehension were applied to assess the validity of the measure of self-perceived oral competence.

In summary, the overall goal of this study was to design, develop and validate an instrument to measure self-perception of oral communicative competence, aimed at students in MS. The specific study objectives were as follows: (1) to design and develop an instrument to measure self-perceived oral competence that incorporates elements of an interactive, functional and pragmatic approach to language development; (2) to validate the instrument with MS students; (3) to test the relationship between the self-perceived oral competence, self-perceived EI and metacomprehension of written texts.

## 2. Methods

### 2.1. Participants

Two samples were used: a first sample in the instrument development phase, consisting of 168 students (51.6% girls) with a mean age of 12.47 years (SD = 0.41), and a second sample of 960 students (52.8% girls) with a mean age of 14.11 years (SD = 0.97). The first pilot sample, with which an initial analysis of internal consistency reliability was carried out, comprised students in their first year of SE, while the second sample, in which the factor structure of the instrument was tested, consisted of students in their first to third year of MS (Middle School in Spain (*Educación Secundaria Obligatoria*, ESO) and comprised four academic years, which were normally taken between the ages of 12 and 16); First year *n*_1_ = 352, second year *n*_2_ = 253 and third year *n*_3_ = 355). Although we used convenience sampling, we attempted to increase the representativeness by recruiting both the first and second samples from State schools located in two regions with different language characteristics: Catalonia, a bilingual Catalan–Spanish community (48.6% first sample, 46.9% second sample), and Madrid, a monolingual Spanish community (51.4% first sample, 53.1% second sample). The instrument was administered exclusively in Spanish.

### 2.2. Instruments

#### 2.2.1. Test of Self-Perceived Oral Competence (TSOC)

Based on the proposed model of oral communication, we developed the initial version of the TSOC, which consisted of 30 indicators or items scored using a 7-point Likert scale that enabled us to sample five content domains or dimensions: (1) interaction management (e.g., When others speak, I pay attention to what they say); (2) multimodality (e.g., Other people clearly understand the emotions I express with my face) and prosody (e.g., I am aware of how my tone of voice and volume can affect others); (3) textual coherence (e.g., I think about the order of the things I am going to say before I speak) and cohesion (e.g., I use expressions or phrases that mark the end of my speech); (4) argumentative strategies (e.g., At the end of my speech, I summarize the most important things I have said); and (5) lexicon and terminology (e.g., In my speech, I use new words I have recently learned). Regarding the differences and similarities between the TSOC and the SSS [22], the former dimension presents the items aimed at a school level, specifically mentioning these deals with classroom-based oral expression. It presents this more concretely and contextually with vocabulary adapted to the comprehension of adolescents. The formulation of the items of the SSS is more abstract and more based on the domination of the message than on intersubjectivity and other skills related to the speaker.

#### 2.2.2. Metacognitive Awareness of Reading Strategies Inventory (MARSI_R,)

We administered the revised version of the Metacognitive Reading Strategies Inventory (MARSI_R, [70]). This is a self-report instrument designed to assess the degree of students’ knowledge or awareness (metacognition) of the strategies they apply when reading school materials used for comprehension. The test consists of 15 items (reading strategies) scored using a 5-point Likert scale (degree to which the student considers that he/she uses the strategy) and grouped into three dimensions: global reading strategies (GRS), problem solving strategies (PPS) and support reading strategies (SRS). The MARSI_R is a good predictor of reading ability level, shows satisfactory factorial validity and internal consistency (Cronbach’s alpha = 0.85), with a subscale reliability of GRS = 0.703, PPS = 0.693 and SRS = 0.743.

#### 2.2.3. Self-Report of Emotional Intelligence (Trait Meta-Mood Scale TMMS-24,)

The TMMS [71] was designed by Salovey et al. [72] to assess how people reflect on their moods and the extent to which they pay attention to their feelings (attention), as well as how they can distinguish (clarity) and self-regulate their feelings to avoid or correct negative moods (repair). The scale is scored using a 5-point Likert scale, with response options ranging from 1 = strongly disagree to 5 = strongly agree. In this study, we used the short version in Spanish [71,73], which contains 24 items and shows an acceptable construct validity, a high internal consistency (Cronbach’s alpha for attention = 0.90, clarity = 0.90 and repair = 0.86) and a satisfactory test–retest reliability, obtaining values ranging from 0.60 to 0.83.

### 2.3. Procedure

Drawing on the theoretical model on which the TSOC is based [10,35], a first version of items was developed and assessed by a panel of 10 experts: two speech therapists, three university teachers conducting research in the field of psycholinguistics and five university teachers of linguistics. The experts were sent a document containing the 30 indicators grouped into content dimensions with the wording proposed by the researchers for each indicator. For each draft proposal they were asked: (1) whether the question was suitable (yes/no); (2) whether the wording was suitable (yes/no); and (3) to give comments/alternative wording. The content validity index calculated from the experts’ degree of agreement about whether the items were suitable for the content being sampled was very high, indicating a 98.52% agreement, and above 88% for all items. There was less consensus regarding the correct wording of the items, with an average of 78.89%. Subsequently, two researchers produced a table from the 10 experts’ responses to the third question, i.e., their proposals for changes to the wording. The options were discussed, and the wording was modified following a process of inter-expert discussion. The initial instrument consisted of 30 items that were administered to the first sample of students in order to assess the psychometric properties of the items. As a result of this initial analysis, 22 items that showed satisfactory discriminant powers were retained, and subsequently, with a second sample, we analyzed the reliability and construct validity of the measure in terms of its factorial structure and the relationship with other relevant variables.

### 2.4. Data Analysis

For the first sample, we analyzed the descriptive statistics of the items and the scale, as well as the corrected total correlation and the internal consistency reliability using Cronbach’s alpha coefficient. For the second sample, goodness of fit was assessed by confirmatory factor analysis (CFA) of the five-factor model and the unidimensional and essential unidimensionality models, assessing the possible effects on the measure resulting from gender and age. We also analyzed correlations between the TSOC scores and the TMMS-24 and MARSI_R. Analyses were performed using SPSS version 25 in the R environment (version 4.1.0) using the *Lavaan* package [74].

## 3. Results

### 3.1. Psychometric Properties of the Items and Reliability of the TSOC

The first objective in constructing the TSOC was to obtain a scale with acceptable internal consistency reliability. Thus, reliability was assessed for the first sample using Cronbach’s alpha coefficient, obtaining a value of 0.80 for the 30 items of the TSOC. In order to maximize instrument reliability, item–test correlations were assessed and items that did not contribute to the internal consistency of the measure were progressively eliminated. After the eliminating eight items, we obtained a shortened version (see Appendix A) of 22 items (score range 22–154 points) with a reliability of 0.85 (see Table 1). This reliability analysis was equivalent to a one-factor exploratory factor analysis (EFA), whereby items with low factor weights were eliminated, and it represents a suitable strategy for achieving measures in which the first factor captures most of the variance [75].

### 3.2. Construct Validity

Although the strategy employed in the initial study ensured the selection of items with the highest weights in the first eigenvalue in order to maximize internal consistency, the theoretical proposal for construct assessment was based on the measurement of five content domains. Thus, in order to validate the factor structure and test whether the proposed theoretical model fit the empirical evidence, we performed a confirmatory factor analysis (CFA) with DWLS estimation [76]. The CFA indicated that the goodness-of-fit to the five-factor model was acceptable, as shown by the following goodness-of-fit indices: χ^2^ (199) = 927.495, *p* < 0.001, *Comparative Fit Index* (CFI) = 0.975, *Tucker–Lewis Index* (TLI) = 0.971, *Root Mean Square Error of Approximation* (RMSEA) = 0.067 (IC 90%: 0.063, 0.072), *Standardized Root Mean Square* (SRMS) = 0.058. We concluded the fit to the five-factor model by applying commonly accepted assessment guidelines which indicate that for a fit to be considered acceptable, CFI and TLI ≥ 0.95 and RMSEA ≤ 0.06 and SRMS ≤ 0.08 [77,78].

Figure 1 shows that the factor weights were acceptable and the correlation between the factors was high, indicating that the instrument presented the property of essential unidimensionality, thus enabling the overall score, rather than only the specific factors, to be used as a measure of oral competence [79,80]. To test for essential unidimensionality, a bifactor model was applied, which showed an excellent goodness of fit: χ^2^ (187) = 450.082, *p* < 0.001, CFI = 0.991, TLI = 0.989, RMSEA = 0.042 (0.037, 0.047) and SRMS = 0.042. We also obtained the following associated indices: explained common variance (ECV) = 0.64, PUC = 0.83, total omega ω = 0.92 and hierarchical omega ω_H_ = 0.87. Consequently, in line with the recommendation of Rodríguez et al. [79] to consider a measure essentially unidimensional when LCA > 0.60 and ω_H_ > 0.70, provided that the percentage of uncontaminated correlations or PUC is greater than 0.80, we can state that the structure is essentially unidimensional. As for the reliability of the overall TSOC score, since the goodness of fit to strict unidimensional structure was rejected (χ^2^ (231) = 2244.080, *p* < 0.001, CFI = 0.929, TLI = 0.922, RMSEA = 0.110 (0.106, 0.114) and SRMS = 0.084), the hierarchical omega = 0.87, rather than total omega, should be selected as the best estimator of reliability, an estimate that is similar to the Cronbach’s alpha coefficient of 0.88.

To assess the possible effect of age and gender, an ANOVA was carried out of the gender variable on the total score of the TSOC, taking age as a covariable. It shows a greater score for girls (M = 93.48, SD = 21.84) compared to boys (M = 89.85, SD = 20.00) which was statistically significant F (1731) = 5.530, *p* = 0.019, η^2^ _partial_ = 0.08; the age covariable was not statistically significant F (1731) = 0.623, *p* = 0.430, η^2^ _partial_ = 0.01.

### 3.3. Construct Validity: Correlations with Other Variables

We analyzed patterns of TSOC correlation with measures of Emotional Intelligence (EI) and metacomprehension. Table 2 shows that the TSOC presented a high correlation with self-perceived EI (attention, clarity and repair) (TMMS-24) and with metacomprehension (GRS, PPS and SRS) (MARSI_r).

We found a statistically significant correlation between the TSOC and the measures of EI and metacomprehension, and homogeneously with all the factors of both measures, indicating that oral competence would be relevant in all the dimensions assessed for EI and metacomprehension.

In short, in terms of validity, the results showed that the TSOC presented a good reliability (between 0.85 and 0.88). We obtained an excellent fit to the five-factor theoretical factor structure, and, in terms of evidence of construct validity in relation to other variables, the TSOC correlated with emotional intelligence, assessed by means of the TMMS-24, and with metacomprehension assessed with the MARSI_R.

## 4. Discussion

In a literature review of research on oral language in MS, Wurth et al. [81] found that, despite the long tradition of teaching public speaking in education, little research attention was paid to oral competence in MS, and few studies examined the practical or affective aspects of oral language teaching in formal contexts. The above study and some of the others reviewed (e.g., [82,83]) highlighted the need to gain a better understanding of how spoken language is being taught and learnt at this stage of education.

Although current legislation is aimed at improving students’ oral communicative skills (Official State Gazette, Organic Law 8/2013) [25], theoreticians and professionals identified inconsistencies in the implementation of this legislation in practice, generated by a mismatch between oral communication teaching in the classroom and educational models that encourage students to remain silent during class [19,84].

To facilitate the development of oral expression as envisaged in the SE curriculum, programs were developed such as the Teaching Program for the Development of Oral Competence [85]. However, despite forming part of the official curriculum, oral expression is rarely explicitly included in classroom teaching activities [86].

Furthermore, oral competence is neither assessed, nor is it included in PISA *(Program for International Student Assessment*) tests to assess school skills [87]. Thus, there is an evident need for effective assessment instruments in SE, in order to develop educational strategies tailored to students’ baseline level of competence [88].

The development of the instrument presented here was based on a theoretical proposal which identified five dimensions of oral communication: (1) interaction management [37,38,89]; (2) multimodality and prosody, comprising two sub-dimensions [37,42,43,45]; (3) textual coherence and cohesion [46,47,48,50,51]; (4) strategies [39,56,57]; and (5) lexicon and terminology [58].

As explained in detail earlier, in order to endow the instrument with a strong content validity, we invited a group of experts to help us devise suitable indicators/items that would measure each of the theoretical dimensions.

Initial analyses showed that although instrument reliability was acceptable, some items presented very low item–test correlations, and this reliability increased to 0.85 once these items were eliminated. In a second, larger and more heterogeneous sample, reliability increased to 0.88, which was a satisfactory level. The factor structure of the instrument in the second sample showed an acceptable factor validity in that it correctly recovered the structure of five correlated factors with an acceptable goodness of fit. To determine whether the structure could be considered essentially unidimensional, alternative unidimensional and bifactor models were tested and the results indicated that the total test score could legitimately be used as a measure of self-perceived oral competence [79,80]. This finding is crucial in that it legitimizes the use of the overall TSOC score, rather than having to make multidimensional use of the instrument by subscale or dimension.

The results on the factorial structure have important implications for the possible uses of the instrument. Although the structure of five correlated structures showed an appropriate goodness of fit (see Figure 1), a hierarchical structure was also observed, in which the factors formed part of a general self-perceived skill (bifactor model). This implies that the items have a relevant double source of variability: first they contribute to the general factor that is to be measured; and second, to the specific factor to which the measure corresponds. As a result, both the use of an overall score, and also the use of scores that the factors, for example, identifying weaknesses and strengths in specific dimensions or assessing how these dimensions develop, were legitimated.

For the study of the convergent validity of the TSOC, theoretically related self-informed measures were taken as a reference, choosing self-efficiency measures instead of measures of competence or performance, as the prior investigation only found moderate correlations between them at 0.30 [90,91]. However, in the future development of the instrument we considered that it was worth exploring to what extent oral competence was related to the perceived oral skill. The aim of this instrument is not to replace the measures of oral competence but to provide a new tool that allows us to assess communicative self-sufficiency and which, as we have shown, is related with variables as relevant in the educational environment, such as EI or reading meta-comprehension.

We believe that the relationship between the TSOC scores and EI and metacomprehension not only provides the instrument with good evidence of construct validity, but also enables us to theorize about the relationships that exist between these three important variables for academic performance. The study by Wurth et al. [81] introduced the possibility of determining the existence of causal relationships between regular oral language practice and self-confidence [82,88,92,93,94]. In our study, we also found significant correlations between competences traditionally associated with oral language, such as emotional intelligence and the self-regulation of reading comprehension. The results also indicate that the measurement of TSOC was shown to be stable for the different ages. This should be analyzed in more depth, given that we can expect its self-effectiveness to improve as its effectiveness increases, although it could also be interpreted as an indicator that students are more aware of their competences and their limitations. In addition, the results indicate that there is a higher score for female students, which is consistent with their greater competence in speech production ([95] and academic performance [96]).

## 5. Conclusions

We obtained very encouraging results for the psychometric properties and validity of the TSOC, yielding an instrument for assessing self-perceived reading skills. However, before these results can be generalized, further research is required with other samples, age groups and cultures, to also test if the TSOC score is stable, or if it varies in formal vs. informal context, or the relative skills of peers, interlocutors, adults... Future directions include investigating how intervention in oral competence impacts on or interacts with these variables and how these relate to academic performance.

## Figures and Tables

**Figure 1 children-08-01136-f001:**
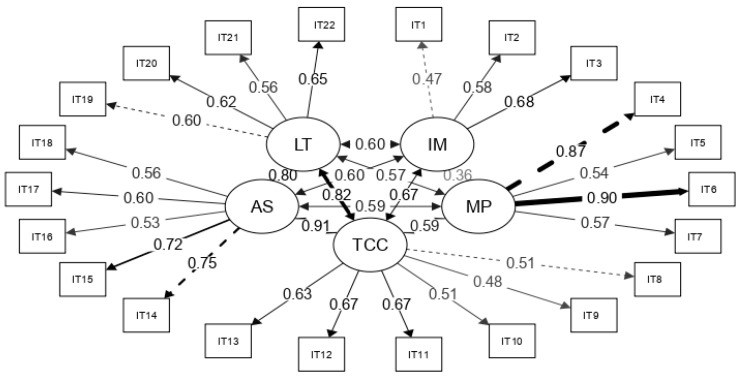
Path diagram of the five-factor model of oral skill. Note: The boxes represent observed variables (items) and the circles latent variables (IM = interaction management, MP = multimodality and prosody, TCC = textual coherence and cohesion, AS = argumentative strategies, and LT = lexicon and terminology). The arrows are the paths with their standardized weights, the double-headed arrows are the correlations between latent variables, the discontinuous lines specify the items that set the metrics and a greater thickness of lines indicates greater weights or correlations.

**Table 1 children-08-01136-t001:** Statistics for the items comprising the final version of the TSOC.

Item	Scale Mean	Scale SD	Total Correlation Item-Test	Cronbach’s Alpha
IT 1	88.15	18.48	0.360	0.848
IT 2	86.91	18.79	0.238	0.852
IT 3	86.50	18.84	0.332	0.848
IT 4	88.43	18.63	0.334	0.849
IT 5	87.91	18.63	0.335	0.849
IT 6	88.45	18.46	0.439	0.845
IT 7	88.19	18.54	0.417	0.845
IT 8	87.15	18.70	0.322	0.849
IT 9	88.07	18.60	0.323	0.849
IT 10	88.61	18.31	0.474	0.843
IT 11	87.98	18.21	0.615	0.838
IT 12	87.93	18.34	0.473	0.843
IT 13	87.87	18.39	0.444	0.844
IT 14	86.61	18.37	0.601	0.840
IT 15	86.92	18.43	0.516	0.842
IT 16	88.62	18.53	0.378	0.847
IT 17	88.91	18.42	0.452	0.844
IT 18	88.73	18.49	0.443	0.845
IT 19	88.85	18.49	0.425	0.845
IT 20	87.40	18.37	0.533	0.841
IT 21	87.74	18.46	0.386	0.847
IT 22	88.17	18.40	0.450	0.844

Note: Means, SD, item-test correlation and Cronbach’s alpha are calculated by eliminating the item analyzed from the scale.

**Table 2 children-08-01136-t002:** TSOC Correlations with the TMMS-24 and MARSI_R.

Variables	1	2	3	4	5	6	7	8	9
**TSOC (1)**	1	0.351 *	0.390 *	0.374 *	0.483 *	0.388 *	0.351 *	0.412 *	0.441 *
Attention (2)		1	0.357 *	0.285 *	0.731 *	0.250 *	0.168 *	0.239 *	0.256 *
Clarity (3)			1	0.482 *	0.799 *	0.178 *	0.140 *	0.135 *	0.186 *
Repair (4)				1	0.768 *	0.211 *	0.138 *	0.138 *	0.189 *
**TMMS-24 (5)**					1	0.295 *	0.201 *	0.225 *	0.288 *
GRS (6)						1	0.586 *	0.586 *	0.836 *
PPS (7)							1	0.671 *	0.866 *
SRS (8)								1	0.880 *
**MARSI_R (9)**									1

* *p* < 0.01 (two-tailed). GRS: global reading; PPS: problem solving; SRS: support reading.

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
