# Peer review of "Assessment of Oral Skills in Adolescents"

_children, 2021, doi:10.3390/children8121136_

Round 1

Reviewer 1 Report

The present manuscript describes a five-dimensional model of oral skills and the development of a self-report questionnaire to assess oral skills in middle schoolers (Test of Self-perceived Oral Competence, TSOC).

While I agree that this is interesting and important, what dampens my enthusiasm somewhat are mainly the following two points:

  1. Oral competence is constructed as multi-dimensional on three levels (factor, dimensions, sub-dimensions), but then collapsed to a unidimensional construct – how does this help us further our understanding of oral competence, which you stress at several points? What I would find more convincing would be a study that demonstrates that the new measure identifies strengths and weaknesses in specific dimensions (such as prosody or cohesion), and thus can help foster oral skills adaptively according to adolescents’ needs (and, as the authors propose, also increases emotional and reading skills, p. 4, ll. 154-158).
  2. As validation, the authors investigated associations with other, converging constructs (EI via TMMS, metacognition on reading strategies via MARSI_R), yet not with other measures of oral skills – this does not provide us with an understanding if and how the new measure is better than existing ones

Perhaps you can address these points in a revision, otherwise I remain unsure if the manuscript actually provides new insights.

Other points and recommendations

  • You emphasise the instrument is theoretically grounded, but because of your brevity I find that hard to follow. To me, it is not fully understandable how the five dimensions were derived, aside from four studies that have investigated oral language teaching (p. 3, ll. 119-128). Can you be more specific?
  • Why use EI, especially the TMMS, which concern intrapersonal “meta-mood” aspects of emotion rather than interpersonal ones?
    • The literature you cited employed other measures of EI that, based on differing models, incorporate such interpersonal scales (e.g. the EQ-i). I suspect that part of the associations between TSOC, TMMS and MARSI_R might be due to general self-efficacy that is applied to oral, emotional, and reading contexts.
  • Why not choose Demir’s SSS, and/or the questionnaire by Campos et al., and demonstrate that the TSOC has better or incremental validity when explaining oral competence?
    • Reading sample items under 2.2.1, how are SSS and TSOC that much different?
  • Assessing competence via self-report is susceptible to biases and does not necessarily reflect actual competence. A more objective measure could help, but that is a very broad criticism and I understand if you would not address that.
  • Methods: were effects of sex/gender, age controlled? Academic achievement/grades?
  • p. 3, l. 119 “In an attempt to develop a less intuitive … instrument”, I’m not sure I understand what you are trying to say. Do you mean less susceptible to social desirability?
  • Table 1: I find the scale means and variances hard to understand, could you indicate if sum scores were calculated (looks like it)?
    • How can variance be that large on a 30-item 7-pt scale?
    • Is the item list available somewhere? Could you indicate to which dimension each item is assigned instead of just terming them “ITEM x”? You might also want to indicate which of these items were cut from the final version (e.g. in a table note).
      • Edit: I now see that items are denoted in Figure 1, but they are hard to decipher
  • Chi² statistics are not well suited when N is large, as your tests turn out significant
  • p. 7 l. 292: why are there two parameters in brackets after the RMSEA? Or do they belong to the SRMS?
  • I miss some descriptive statistics that could give the reader a grasp on how well-developed oral skills actually were. How high were scores on the TMMS and MARSI_R?
    • Because self-report measures tend to score higher than objective ones, especially if your sample has a high self-efficacy, it might influence your results.
  • Figure 1: please indicate what different line styles mean (bold, dashed, ...), and explain one- and double-headed arrow for readers who are not familiar with SEM or CFA
  • As I said, I think it’s somewhat a shame that the TSOC is collapsed to a single score – I imagine it would be more interesting to show how TMMS and MARSI_R correlate with single dimensions of oral skills
  • Table 2: there is an error in enumeration in the head row; also, I think you are missing an asterisk at *p < .01
  • The reference list is incomplete (see 7.-11.)
  • An additional idea: would it be interesting to test if the TSOC score is stable, or if it varies in formal vs. informal context, or the relative skills of peers, interlocutors, adults?

Author Response

While I agree that this is interesting and important, what dampens my enthusiasm somewhat are mainly the following two points:

1.Oral competence is constructed as multi-dimensional on three levels (factor, dimensions, sub-dimensions), but then collapsed to a unidimensional construct – how does this help us further our understanding of oral competence, which you stress at several points? What I would find more convincing would be a study that demonstrates that the new measure identifies strengths and weaknesses in specific dimensions (such as prosody or cohesion), and thus can help foster oral skills adaptively according to adolescents’ needs (and, as the authors propose, also increases emotional and reading skills, p. 4, ll. 154-158).

We consider that it is fundamental that the question raised by the reviser is clarified in the manuscript; accordingly we have included the following explanation in the discussion (see lines 421-429):

The results on the factorial structure have important implications on the possible uses of the instrument. Although the structure of five correlated structures showed an appropriate goodness of fit (see Figure 1), a hierarchical structure is in turn observed in which the factors form part of a general self-perceived skill (bifactor model). This implies that the items have a relevant double source of variability: first they contribute to the general factor that is to be measured; and second, to the specific factor to which the measure corresponds. As a result, both the use of an overall score, and also the use of scores of the factors are legitimated for, for example, identifying weaknesses and strengths in specific dimensions or assessing how these dimensions develop.

2. via TMMS, metacognition on reading strategies via MARSI_R), yet not with other measures of oral skills – this does not provide us with an understanding if and how the new measure is better than existing ones

The following explanation has been included in the discussion (see lines 430-438):

 For the study of convergent validity of the TSOC, theoretically related self-informed measures were taken as a reference, choosing self-efficiency measures instead of measures of competence or performance, as the prior investigation has only found moderate correlations between them at 0.30 [[90-91]. However, in the future development of the instrument we consider that it is worth exploring to what extent oral competence is related to the perceived oral skill. The aim of this instrument is not to replace the measures of oral competence but to provide a new tool that allows us to assess communicative self-sufficiency and which, as we have shown, is related with variables as relevant in the educational environment such as IE or reading meta-comprehension.

Other points and recommendations:

You emphasise the instrument is theoretically grounded, but because of your brevity I find that hard to follow. To me, it is not fully understandable how the five dimensions were derived, aside from four studies that have investigated oral language teaching (p. 3, ll. 119-128). Can you be more specific?

In response to the request to go more in-depth in the theoretical framework, we have introduced some ideas that lie at the heart of the preliminary work performed by the team in relation to the analysis of situations of communicative interaction in university classes where future teachers are trained. These contributions have sought to show that the theories of the argument and analysis of the discourse form part of the basis of the instrument in the section that precedes the one described in this article – the TSOC – along with a whole series of research and contributions by different authors that stress the importance of natural interactions between children and their carers of all ages as a key element for the development of language in any modality (oral, through gestures, augmentative systems and sign language). This information has been included in lines 124-187.

Why use EI, especially the TMMS, which concern intrapersonal “meta-mood” aspects of emotion rather than interpersonal ones? The literature you cited employed other measures of EI that, based on differing models, incorporate such interpersonal scales (e.g. the EQ-i). I suspect that part of the associations between TSOC, TMMS and MARSI_R might be due to general self-efficacy that is applied to oral, emotional, and reading contexts.

The question raised by the reviser is very interesting to explore; we agree with the reviser that part of the variance shared by the three measures used is probably due to the three being self-efficacy measures, but this is a key question for obtaining converging evidence, since the measures must refer to theoretically related constructs or measures. At any event, we feel that obtaining discriminatory evidence is also important, and hence take note of the reviser’s recommendation for any future research we undertake.

Why not choose Demir’s SSS, and/or the questionnaire by Campos et al., and demonstrate that the TSOC has better or incremental validity when explaining oral competence? Reading sample items under 2.2.1, how are SSS and TSOC that much different?

We do not consider it appropriate to compare the TSOC with these tests that served as a reference and, in fact, some of the items are very similar, and hence necessarily, for their own benefit, are related. However, unlike these instruments, the TSOC was drafted based on the identification of basic communicative dimensions, although as previously remarked, a general factor exists that underlies the different measures of self-perceived communication skills. The TSOC allows an evaluation to be made of specific factors, as indicated in the new paragraph of the discussion.

As regards the differences and similarities between the TSOC and the SSS, the former presents the items aimed at a school level, specifically mentioning that that these deals with classroom-based oral expression. It does this more concretely and contextually with vocabulary adapted to the comprehension of adolescents. The formulation of the items of the SSS is more abstract and more based on the domination of the message than on intersubjectivity and other skills related to the speaker. For example:

SSS: I pause in the right places.

TSOC: You take pauses so that what you say can be better followed.

SSS: I use body language effectively in my speech.

TSOC: When you address the whole class, you don’t know what to do with your hands. 

Assessing competence via self-report is susceptible to biases and does not necessarily reflect actual competence. A more objective measure could help, but that is a very broad criticism and I understand if you would not address that.

A paragraph has been included in the discussion to this end (see lines 430-438):

Methods: were effects of sex/gender, age controlled? Academic achievement/grades?

To assess the potential effect of age and gender, ANOVA was performed of the gender and age (covariable) on the total score of the TSOC (see lines 361-365 (results) and 446-452 (discussion)).                         

“In an attempt to develop a less intuitive … instrument”, I’m not sure I understand what you are trying to say. Do you mean less susceptible to social desirability?

We have removed this part of the sentence - “a less intuitive” - from the text so as not to confuse the reader.

Table 1: I find the scale means and variances hard to understand, could you indicate if sum scores were calculated (looks like it)? How can variance be that large on a 30-item 7-pt scale?

Table 1 has been corrected so as to make it more understandable, and the SD is now reported instead of the variance.

Is the item list available somewhere? Could you indicate to which dimension each item is assigned instead of just terming them “ITEM x”? You might also want to indicate which of these items were cut from the final version (e.g. in a table note). … I now see that items are denoted in Figure 1, but they are hard to decipher

The list of all the TCOA items has been included in an appendix at the end of the article.

Chi² statistics are not well suited when Nis large, as your tests turn out significant

Although the observation is right, since it is the only inferential index, its value must be reported since it could be used for comparing between competing models

p. 7 l. 292: why are there two parameters in brackets after the RMSEA? Or do they belong to the SRMS?

The first time this was stated, it was indicated that this refers to the confidence range (IC 90%).

I miss some descriptive statistics that could give the reader a grasp on how well-developed oral skills actually were. How high were scores on the TMMS and MARSI_R? Because self-report measures tend to score higher than objective ones, especially if your sample has a high self-efficacy, it might influence your results.

The average scores of the TMMS24 (Average = 76.13 out of 120 points) and of the MARSI_R (Average = 51.47 out of 75) were not particularly high when compared with the highest scores; however, we agree with the reviser that, in the future, it would be pertinent and important to study the relationship between efficacy and self-efficacy, along with other personality variables that may influence the self-perception of their skills.

 Figure 1: please indicate what different line styles mean (bold, dashed, ...), and explain one- and double-headed arrow for readers who are not familiar with SEM or CFA

An explanatory note has been included (lines 354-359).

As I said, I think it’s somewhat a shame that the TSOC is collapsed to a single score – I imagine it would be more interesting to show how TMMS and MARSI_R correlate with single dimensions of oral skills

As mentioned previously, the total score of the TSOC can be legitimately used to evaluate its correlation with other variables of interest (e.g. TMMS and MARSI_R) by adjusting its factorial structure to a bifactor model, and it is also possible to study the correlation patterns of specific factors should this be of interest to the applied researcher.

Table 2: there is an error in enumeration in the head row; also, I think you are missing an asterisk at *< .01

We have corrected Table 2.

The reference list is incomplete (see 7.-11.)

The references have been completed

An additional idea: would it be interesting to test if the TSOC score is stable, or if it varies in formal vs. informal context, or the relative skills of peers, interlocutors, adults?

We consider the reviser’s proposal to be very interesting, which has been included as a future line of research at the end of the discussion.

We wish to thank the reviser for the comments and suggestions, which have been very useful in improving the manuscript.

Reviewer 2 Report

Thank you for the opportunity to review this very interesting paper. The study developed and applied a new self-rating measure of communication skills for middle-school aged children and tested the measure on a large cohort of young people in Spain (n=960, mean age 14;11). The strength of this work is the novel research area - a self rating scale for adolescents in terms of their own communication skills, particularly in Spain. The large scale of the cohort is also a real strength of the study. 

Further work is needed to:

  • Draw out the significance of this research in the literature review.  At the moment, the literature review is rather difficult to read, and it is hard to decipher the real need for a self-rating tool like this.
  • Argue for the uniqueness of this tool. The tool itself should ideally be provided with the paper - it is only 22 items and I'm not sure why it wasn't included as an appendix. This would significantly increase the impact of the paper.
  • Without the rating tool itself, it is hard to see why this tool would be used over (for example) the Strengths and Difficulties Questionnaire; the Children's Communication Checklist-2; Conversation Participation Rating Scale in the CELF-5 etc. 
  • The self-rating tool is tested in association with self-rating scales of reading strategies and self-rating scale of emotional intelligence. This is set out in the literature review but I was a little confused about the rationale for this. It would have been interesting to see correlations with a measure of communication itself - assuming there is no such data, this should be highlighted as a limitation / area for future work. 
  • Some of the example items on the self-rating tool seemed very difficult to comprehend - e.g. 'At the end of my intervention, I summarise the most important things I have said' - 'intervention' is not clear in this context. This may be a translation issue but it seems important to acknowledge that this tool will only be accessible to those with strong spoken and written language skills?
  • The statistical modelling used seemed sound but this is not an area of expertise and I would recommend a review which examines the results specifically. 

I've highlighted the limitations to the scope of the study but the paper is well written, interesting, and has the potential to make an important contribution to how educators consider communication in middle schools. 

Author Response

Further work is needed to:

  1. Draw out the significance of this research in the literature review.  At the moment, the literature review is rather difficult to read, and it is hard to decipher the real need for a self-rating tool like this.

We have incorporated theoretical elements on language learning in natural contexts and the strategies used by adults, both families and teachers, in these situations, in a natural way to foster the development of language, particularly in conversation (lines 140-144). Elements have also been introduced in the theory of the argument and elements of the methodologies of the analysis of the discourse (lines 125-134)

  1. Argue for the uniqueness of this tool. The tool itself should ideally be provided with the paper - it is only 22 items and I'm not sure why it wasn't included as an appendix. This would significantly increase the impact of the paper. Without the rating tool itself, it is hard to see why this tool would be used over (for example) the Strengths and Difficulties Questionnaire; the Children's Communication Checklist-2; Conversation Participation Rating Scale in the CELF-5 etc. 

Following the reviser’s recommendation, 22 items of the TSOC have been included in the appendix (lines 674-709)

 3.The self-rating tool is tested in association with self-rating scales of reading strategies and self-rating scale of emotional intelligence. This is set out in the literature review but I was a little confused about the rationale for this. It would have been interesting to see correlations with a measure of communication itself - assuming there is no such data, this should be highlighted as a limitation / area for future work. 

This instrument does not seek to replace oral skill measures but offers a new tool on communicative self-efficacy that did not exist (see lines 430-438):

For the study of convergent validity of the TSOC, theoretically related self-informed measures were taken as a reference, choosing self-efficiency measures instead of measures of competence or performance, as the prior investigation has only found moderate correlations between them at 0.30 [90-91]. However, in the future development of the instrument we consider that it is worth exploring to what extent oral competence is related to the perceived oral skill. The aim of this instrument is not to replace the measures of oral competence but to provide a new tool that allows us to assess communicative self-sufficiency and which, as we have shown, is related with variables as relevant in the educational environment such as IE or reading meta-comprehension.

This has been included for future lines of research (lines 456-457)

  1. Some of the example items on the self-rating tool seemed very difficult to comprehend - e.g. 'At the end of my intervention, I summarise the most important things I have said' - 'intervention' is not clear in this context. This may be a translation issue but it seems important to acknowledge that this tool will only be accessible to those with strong spoken and written language skills?

In relation to this comment, it is indeed true that this is due to translation questions in some cases, such as the one indicated by the reviser. When we speak about “intervention”, we are referring to “oral participation in class”, which would rather be “speech participation” or “communicative or linguistic participation”. At any event, some items of the TSOC refer to classroom situations in which pupils habitually participate, such as discussions, conversation, opened up for everyone after working in collaborative groups or work undertaken in projects or individuals presentations of a few minutes in length. Specifically, this item referred to by the reviser seeks for pupils to self-assess their ability to summarise their own spontaneous or previously prepared “communicative participation” in front of their classmates and teacher, in order to facilitate, to their other classmates, and to the teacher, their understanding because it has been long, complex and contained many ideas. This is probably a translation problem, given that the accessibility and complexity of all the items is adequate for the pupils that the test is aimed at. In no case did the participating pupils raise problems as to their comprehension of the items. The translation of these examples has been revised (lines 253-256).     

 5.The statistical modelling used seemed sound but this is not an area of expertise and I would recommend a review which examines the results specifically. 

We have revised the analysis techniques in-depth to ensure they are correct and adequate. English was also checked by the language service of the university.

  1. I've highlighted the limitations to the scope of the study but the paper is well written, interesting, and has the potential to make an important contribution to how educators consider communication in middle schools. 

We wish to thank the reviser for the comments and suggestions, which have been very useful in improving the manuscript.

Round 2

Reviewer 1 Report

I thank the authors for this revision and their responses. I find the manuscript has improved especially by the additions to the theory section and the item list.

While I remain sceptical about the convergent validity with other self-efficacy measures and lack of more objective measures of either communicative, emotional, or meta-comprehension competences, I think I can agree with the authors that future research can look more into these aspects. The authors mention that “prior investigation has only found moderate correlations between [performance measures/competences and self-reports] at 0.30”, which is exactly the point I am trying to raise. While I agree that self-efficacy of students is important, and that for your measure you decided to stick to that, I strongly belief that their actual competences need to be acknowledged as well. After all, students may indicate they feel they are good communicators, but what is the benefit for them when in a situation that demands such skills they deliver below their own expectations?

I still find some minor points and hope the authors are willing to look over their revision once more.

TSOC and SSS

  • You write in your response: “As regards the differences and similarities between the TSOC and the SSS, the former presents the items aimed at a school level, specifically mentioning that that these deals with classroom-based oral expression. It does this more concretely and contextually with vocabulary adapted to the comprehension of adolescents. The formulation of the items of the SSS is more abstract and more based on the domination of the message than on intersubjectivity and other skills related to the speaker.” --> I find this quite important, why not add this explanation to the manuscript, for example in section 2.2.1?

Table 1

  • My apologies, but it is still unclear to me how the means can be around 80 when the scale is anchored from 1 to 7 (or 1 to 5). Seems you calculated sum scores for each variable, and then reported means of such scores across participants. Maybe that is common in the literature you cited, but to me it was strange since I am used to reading means that are calculated immediately without summing up (i.e., TMMS mean score would be something like M = 4.1). I found it really helpful that in your response, you stated a max score of 120 for the TMMS – now it is easier to evaluate a score of ~76. This means for the TSOC the max score would be 154, or am I wrong? You might want to add this info to your manuscript (and the Table 1 note).

SRMS

  • Thank you for the clarification regarding the confidence interval, but there is still the value to the SRMS missing on p. 8 l. 494.

Fix some typos:

  • 4 l. 222: “this basic skills” --> these basic skills, or this basic skill
  • 5 ll. 236-237: “In relation to the link … [60], found that listening comprehension …” --> The sentence is missing a subject
  • 6 ll. 287-289: maybe “speech” instead of “intervention”?
  • Section 3.3: EQ --> EI
  • 10 l. 707: IE --> EI

Author Response

We want to thank the reviewer once again for his careful review. Following your observations, you will see the following changes in blue ink:
- The paragraph indicated by the reviewer has been included at the end of section 2.2.1
- The range of TSOC scores has been included in section 3.1.
- SRMR values have been included (section 3.2).
- The typographical errors detected by the reviewer have been corrected.